# Magnesium Hydroxide Nanoparticles Inhibit the Biofilm Formation of Cariogenic Microorganisms

**DOI:** 10.3390/nano13050864

**Published:** 2023-02-25

**Authors:** Kentaro Okamoto, Daisuke Kudo, Dao Nguyen Duy Phuong, Yoshihito Iwamoto, Koji Watanabe, Yoshie Yoshioka, Wataru Ariyoshi, Ryota Yamasaki

**Affiliations:** 1Division of Infections and Molecular Biology, Department of Health Promotion, Kyushu Dental University, Kitakyushu, Fukuoka 803-8580, Japan; 2Division of Developmental Stomatognathic Function Science, Department of Health Promotion, Kyushu Dental University, Kitakyushu, Fukuoka 803-8580, Japan; 3SETOLAS Holdings Inc., Hayashida-cho, Sakaide, Kagawa 762-0012, Japan; 4Collaborative Research Centre for Green Materials on Environmental Technology, Kyushu Institute of Technology, 1-1 Sensui-chou, Tobata-ku, Kitakyushu, Fukuoka 804-8550, Japan

**Keywords:** magnesium hydroxide nanoparticles, dental caries, cariogenic microorganisms, biofilm inhibition

## Abstract

Although various caries-preventive agents have been developed, dental caries is still a leading global disease, mostly caused by biological factors such as mutans streptococci. Magnesium hydroxide nanoparticles have been reported to exhibit antibacterial effects; however, they are rarely used in oral care practical applications. In this study, we examined the inhibitory effect of magnesium hydroxide nanoparticles on biofilm formation by *Streptococcus mutans* and *Streptococcus sobrinus*—two typical caries-causing bacteria. Three different sizes of magnesium hydroxide nanoparticles (NM80, NM300, and NM700) were studied, all of which inhibited biofilm formation. The results showed that the nanoparticles were important for the inhibitory effect, which was not influenced by pH or the presence of magnesium ions. We also determined that the inhibition process was mainly contact inhibition and that medium (NM300) and large (NM700) sizes were particularly effective in this regard. The findings of our study demonstrate the potential applications of magnesium hydroxide nanoparticles as caries-preventive agents.

## 1. Introduction

Dental caries is one of the most common diseases worldwide and is primarily caused by oral bacteria. In 2019, the *Lancet* published a special issue on oral health [1,2], in which it reported that 34% of people have untreated dental caries [1]. The risk of caries development occurs when teeth erupt into the oral cavity and the surface of the tooth comes into contact with oral bacteria. In other words, dental caries occurs when the hard tissues of teeth (i.e., dentin, cementum, and mostly the enamel) are exposed to the oral environment and are subsequently destroyed by the action of oral bacteria. Early *Streptococcus* spp. colonizers specifically adhere to and settle on the protein receptors of the pellicle––a thin film composed of saliva-derived proteins and glycoproteins––formed on the enamel surface. Bacteria also specifically bind to each other, resulting in an increase in the number of bacterial species in the plaque [3]. Mutans streptococci are considered caries-causing bacteria that form biofilms on tooth surfaces, metabolize sugars, and produce acid, which demineralizes the teeth and causes dental caries [4]. Among mutans streptococci, *Streptococcus mutans* and *Streptococcus sobrinus* cause dental caries in humans [5]. Both bacteria produce insoluble glucans from sucrose via glucosyl transferase, which aids in the formation of a strong biofilm [6]. Currently, the only treatment for dental caries is to scrape off the affected area. Brushing is the most effective way to remove plaque, but some people, such as the elderly and children, have difficulty brushing carefully. Furthermore, unlike wounds on bones or skin, caries cannot be repaired; therefore, prevention of dental caries is the key. In this sense, drugs that prevent dental caries and assist in oral care are needed. However, there is still no general prophylactic agent for dental caries that is recognized worldwide.

Since caries is mainly caused by oral bacteria, eliminating the cariogenic bacteria, such as *S. mutans* and *S. sobrinus*, will reduce the risk of dental caries. The inhibitory effects of many chemicals and materials, such as organic materials, inorganic materials, and natural products from plants, on cariogenic bacteria have been reported. As organic materials, methylene blue and chlorin e6 have been found to reduce the biofilm viability of *S. mutans* [7]. As inorganic materials, nano-silver and nano-zinc oxide have been demonstrated to inhibit *S. mutans* growth [8,9]. As a plant extract, the ethanol extract of *Aralia continentalis* or the natural plant product trans-chalcone have been shown to inhibit *S. mutans* biofilm formation [10,11].

Mouthwash is commonly used as an oral care product, and some formulations contain ethanol to dissolve the active ingredients and provide oral refreshment. However, ethanol is a highly volatile ingredient, and its excessive use can decrease saliva in the mouth. When saliva volume is reduced and the mouth becomes dry, symptoms such as bad breath, tooth decay, and gum disease are more likely to occur. Furthermore, there are reports that mouthwashes containing ethanol increase the risk of oral cancer owing to the acetaldehyde generated [12]. Hence, the use of highly volatile chemicals, such as ethanol, is undesirable. In addition, sodium hypochlorite, which is frequently used in dental treatments, has a pronounced sterilizing effect, but is also known to be harmful to the oral mucosa [13,14]. From these perspectives, there is a need for new agents that minimize the risk of deterioration of the oral environment.

Magnesium hydroxide (Mg(OH)_2_) is used in a variety of medical products, such as antacids and laxatives [15,16,17], because of its low cost and low risk of adverse effects on the human body. Its safety is generally recognized by the FDA (E number E528). In recent years, many studies have reported the antibacterial effects of Mg(OH)_2_ nanoparticles (NPs), not only on bacteria [18,19], but also on fungi and protozoa [20]. We previously constructed three different sizes of Mg(OH)_2_ NPs, namely NM80, NM300, and NM700, and demonstrated their bactericidal effects against *Escherichia coli* [21]. Even in persisters, *E. coli* (bacteria that are highly resistant to drugs and other stresses [22,23,24,25]) were able to be sterilized [21].

In this study, Mg(OH)_2_ NPs were used as new medical materials to treat dental caries and for oral care. To this end, we investigated whether Mg(OH)_2_ NPs have an inhibitory effect on bacterial growth and biofilm formation associated with dental caries.

## 2. Materials and Methods

### 2.1. Bacterial Strains

*S. mutans* UA159 and *S. sobrinus* ATCC 33478 were streaked from glycerol stock onto brain heart infusion (BD, Franklin Lakes, NJ, USA) agar plates containing 1% yeast extract (BHIY; BD) and placed in a CO_2_ incubator (SANYO Electric Co., Ltd., Osaka, Japan) at 37 °C for two days with 5% CO_2_. Afterwards, a single colony was picked from the agar plates, inoculated into BHIY broth, and incubated at 37 °C for 16 h with 5% CO_2_.

### 2.2. Mg(OH)_2_ NP Synthesis

Mg(OH)_2_ NPs (NM80, NM300, and NM700) were synthesized as described in our previous study [21]. Briefly, magnesium chloride and sodium hydroxide were used for NP synthesis. The size of the NPs was adjusted by changing the mixing ratio and the reaction time. These processes yielded NPs with sizes of approximately 80, 300, and 700 nm. The NPs were observed using scanning electron microscopy (S-4300; HITACHI, Tokyo, Japan) (Figure 1). The amount of magnesium ions eluted into the solvent and the pH of the solution were measured via inductively coupled plasma optical emission spectrometry (ICP-OES, SPS3500DD; HITACHI) and a pH meter (LAQUAact D-71; HORIBA, Kyoto, Japan), respectively.

### 2.3. Biofilm Assay, Pre-Treatment for Biofilm Inhibition, and Biofilm Dispersal Assay

BHIY containing 0.1% sucrose (Sigma Aldrich, St. Louis, MO, USA) for *S. mutans* and 0.2% sucrose for *S. sobrinus* were prepared. Next, each type of the Mg(OH)_2_ NPs were added at a final concentration of 1000 mg/L. These solutions were diluted twice in a 96-well microtiter plate (Iwaki; AGC Techno Glass Co., Ltd., Shizuoka, Japan), and each overnight bacterial culture was added at a turbidity of 0.05 at 600 nm. Prepared sample plates were incubated at 37 °C for 24 h with 5% CO_2_. Then, absorbance was measured at 620 nm for growth inhibition analysis.

The biofilm assay was performed as previously described [26]. Briefly, the sample plate was prepared using the method described above. The supernatant in the 96-well microtiter plate was discarded after 24 h of incubation, and the biofilm plate was washed with distilled water three times. Biofilms were stained using 0.1% crystal violet for 20 min at 25 °C, and stained biofilms were dissolved in 95% ethanol. The total amount of biofilm formed was measured at an absorbance of 540 nm.

As a pre-treatment method for attachment inhibition, Mg(OH)_2_ NPs in phosphate-buffered saline (PBS) solution (1000 mg/L) were added to the microtiter plate, and incubation was conducted at 37 °C for 24 h. Next, supernatants were discarded, and bacterial cultures (turbidity of 0.05 at 600 nm) were added to the pre-treated microtiter plate, which was followed by incubation at 37 °C for 24 h with 5% CO_2_.

For biofilm dispersal, *S. mutans* and *S. sobrinus* were incubated in microtiter plates at 37 °C for 24 h with 5% CO_2_, after which the supernatants were discarded and washed. Mg(OH)_2_ NPs in PBS solution (1000 mg/L) were added to the biofilm plate. The plates were shaken at 2500 rpm for 1 min. In both pre-treatment and dispersal methods, PBS was used as the control. Both dispersal and pre-treatment methods after incubation and biofilm assays were performed using the same method described above.

## 3. Results

### 3.1. S. mutans and S. sobrinus Growth and Biofilm Inhibition by NM80, NM300, and NM70

The growth and biofilm formation inhibitory effects of Mg(OH)_2_ NPs on *S. mutans* and *S. sobrinus* are shown in Figure 2. As can be seen in the results of the growth inhibition assay of both strains, there were no inhibitory effects on growth at any concentration (0–1000 mg/L) (Figure 2, green bars). In contrast, biofilm inhibition was observed. All Mg(OH)_2_ NPs significantly inhibited *S. mutans* at concentrations of 250 mg/L or more and *S. sobrinus* at concentrations of 500 mg/L or more (Figure 2, blue bars). The highest biofilm inhibitory effect was observed at 1000 mg/L for all Mg(OH)_2_ NPs. NM80, NM300, and NM700 inhibited *S. mutans* biofilms by 54%, 75%, and 87%, respectively. For *S. sobrinus*, NM80, NM300, and NM700 inhibited biofilms by 66%, 73%, and 80%, respectively. These results show that Mg(OH)_2_ NPs cannot effectively inhibit bacterial growth but can inhibit biofilm formation.

### 3.2. Influence of pH and Magnesium Ions on Biofilm Inhibition

Since Mg(OH)_2_ NPs may change the pH of the solution, the pH of Mg(OH)_2_ NPs dissolved in BHIY broth was measured. The pH values of NM80, NM300, and NM700 at a concentration of 1000 mg/L were 8.56 ± 0.02, 7.6 ± 0.2, and 7.38 ± 0.09, respectively. NM300 and NM700 were nearly neutral, whereas NM80 was slightly basic. Therefore, we examined whether biofilm inhibition was affected by changes in pH. Biofilms in culture medium with a pH of 8.5 were measured and compared to that in 1000 mg/L of each type of Mg(OH)_2_ NPs. At a pH of 8.5, there was a slight inhibitory effect compared with that observed at a pH of 7 for both *S. mutans* and *S. sobrinus*, but the effect was small compared to the inhibition caused by the Mg(OH)_2_ NPs (Figure 3A). Therefore, it is clear that the inhibitory effect was mainly due to the Mg(OH)_2_ NPs rather than the change in pH.

Next, the biofilm inhibitory effect of magnesium ions was confirmed because some magnesium ions were eluted from the Mg(OH)_2_ NPs [21]. Magnesium sulfate was used instead of Mg(OH)_2_ NPs at a concentration of 0–1000 mg/L to examine the biofilm inhibition effects on *S. mutans* and *S. sobrinus*. No inhibitory effect on the biofilm was observed, indicating that magnesium ions had no influence on biofilm formation (Figure 3B). Growth was also not influenced by magnesium ions in either *S. mutans* or *S. sobrinus* (Appendix A). Hence, it is clear that Mg(OH)_2_ NPs, and not magnesium ions, mediated the inhibition of biofilm formation.

### 3.3. Biofilm Inhibition by Pre-Treating Mg(OH)_2_ NPs

The effect of biofilm inhibition by Mg(OH)_2_ NPs is shown in Figure 2. First, attachment inhibition by pre-treating the plates with each type of Mg(OH)_2_ NPs (1000 mg/L) was confirmed. Compared to that with pre-treatment with PBS alone as a control, NM300 and NM700 significantly reduced biofilm formation in both *S. mutans* and *S. sobrinus*, whereas NM80 showed no inhibitory effect (Figure 4A). The reduction was 50% for NM300 and 45% for NM700 in *S. mutans* and 75% for NM300 and 73% for NM700 in *S. sobrinus*. An inhibitory effect on growth was not observed after pre-treatment with Mg(OH)_2_ NPs (Appendix A).

Next, we examined whether Mg(OH)_2_ NPs could disperse matured biofilms. As one application of NPs, we considered a mouthwash. Assuming the medicinal effect in the oral cavity, a Mg(OH)_2_ NP solution was added to the formed biofilm and dispersal treatment at 2500 rpm for 1 min was performed. However, no dispersion-removal effect was observed for each type of Mg(OH)_2_ NPs (Figure 4B). Rather, there was an increase in apparent biofilm size compared to the untreated (Ctr) group for both strains. Hence, the inhibition point is not after maturation, as the effect on mature biofilms was found to be poor.

## 4. Discussion

Medicinal treatments with potential antimicrobial efficacy against oral pathogenic bacteria are being studied and developed. This study proposes Mg(OH)_2_ NPs as a new material expected to have an anti-biofilm effect against mutans streptococci, a group of bacteria that cause dental caries. Three different sizes of Mg(OH)_2_ NPs (NM80, NM300, and NM800) were used and their inhibitory effects on growth and biofilm formation were demonstrated for *S. mutans* and *S. sobrinus*. Although no Mg(OH)_2_ NPs showed an inhibitory effect on growth, an inhibitory effect on biofilm formation was observed, with the greatest inhibition observed at a concentration of 1000 mg/L (Figure 2). Previous studies have reported that Mg(OH)_2_ NPs kill *E. coli* [18,21]; however, Mg(OH)_2_ NPs did not inhibit bacterial growth or kill bacteria in our study. *S. mutans* and *S. sobrinus* are smaller than *E. coli* and have a cocci shape rather than the bacillus shape of *E. coli*. This may have caused a significant difference in the bactericidal (growth inhibitory) activity. With regard to the size of Mg(OH)_2_ NPs, the larger the size, the more effective the inhibition of biofilm formation. NM80, NM300, and NM700 inhibited *S. mutans* biofilms by 54%, 75%, and 87%, respectively. NM80, NM300, and NM700 inhibited *S. sobrinus* biofilm formation by 66%, 73%, and 80%, respectively. (Figure 2). The effect of these Mg(OH)_2_ NPs was not efficient in terms of their pH or magnesium ions, as both were slightly inhibited by mutans streptococci biofilms (Figure 3). Biofilm formation mainly includes adhesion, colonization, proliferation, and maturation [27]. To investigate which processes are affected by the inhibitory effect of Mg(OH)_2_ NPs, the effect of Mg(OH)_2_ NP pre-treatment on adhesion inhibition or dispersion into the biofilm after maturation was examined, as shown in Figure 4. A comparison of the attachment inhibition by pre-treatment revealed that NM300 and NM700 strongly inhibited (reduced 45–50% against *S. mutans* and 73–75% against *S. sobrinus*, Figure 4A). Based on these results, it can be concluded that the larger the particle size, the more effective the biofilm inhibition. Larger Mg(OH)_2_ NPs tend to sink to the bottom, which may have prevented the bacteria from adhering to the bottom. The smaller NPs (NM80) did not inhibit the attachment and maturation processes (Figure 4); therefore, we presumed that NM80 primarily prevented the proliferation step after adhesion, and NM300 and NM700 primarily prevented the attachment process. However, their dispersing effect on mature biofilms was poor, resulting in an apparent increase in biofilms (Figure 4B). This has not been verified, but this may be due to the deposition of Mg(OH)_2_ NPs on the mature biofilm. In this study, we used three different sizes (80, 300, and 700 nm) of Mg(OH)_2_ NPs for biofilm inhibition testing. The larger size had a better inhibition effect; therefore, larger sizes (1000 nm or more) may show better results. In a future study, larger-sized Mg(OH)_2_ NPs should be tested.

Recently, a study reported that oral flora balance is related to oral or human body health and diseases [28]. In addition, the plaque flora plays a major role in the risk of dental caries. Plaques with a high percentage of cariogenic bacteria, such as *S. mutans*, contain more acid-retentive glucans, which increases the acidity level (decreasing plaque pH) [29]. This creates an environment in which acid-tolerant *S. mutans* are more likely to proliferate, creating a vicious cycle, resulting in caries formation [30]. In oral care, drugs that unnecessarily sterilize the oral flora might instead upset the balance of the oral flora and negatively affect the environment. Mg(OH)_2_ NPs have been shown to inhibit the adhesion of mutans streptococci and biofilm formation, although they did not show growth-inhibitory effects. It is very important to note that the results do not suggest an effect on the oral microflora, but indicate that they can inhibit biofilms, which are the main cause of dental caries. In other words, Mg(OH)_2_ NPs can reduce the risk of dental caries without compromising the balance of the oral environment.

This study has some limitations. Numerous papers related to metal nanoparticles, such as silver, gold, copper, titanium, and zinc, have reported antibacterial effects [31,32,33,34]. This is because bacteria are much less likely to acquire resistance to metal nanoparticles than to other conventional narrowly targeted antibiotics [35]. It has also been reported that Ag-, Au-, or Pd-doped apatite suppresses the biofilm formation of drug-resistant bacteria (methicillin-resistant *Staphylococcus aureus* and vancomycin-resistant *Enterococcus faecalis*) [36], making the use of antimicrobial metals as novel antimicrobial materials an important position. There are also various mechanisms of nanoparticle disinfection, including generation of reactive oxygen species (ROS), biomolecular interactions and regulation, ATP depletion, and membrane interactions [32,34]. It was revealed that Mg(OH)_2_ NPs could inhibit biofilms of mutans streptococci. In particular, NM300 and NM700 inhibited the attachment process of biofilm formation (Figure 4A). However, we were unable identify the underlying mechanisms by which Mg(OH)_2_ NPs inhibit biofilm formation (chemically, physically, or other). As a biofilm inhibition mechanism, it has been reported In various bacterial biofilms, such as that of *Pseudomonas aeruginosa*, *Clostridioides difficile*, and *Campylobacter jejuni*, that cyclic diguanylate (c-di-GMP) is related to biofilm properties [37,38,39,40]. Although we did not examine the genetic mechanism of biofilm inhibition by Mg(OH)_2_ NPs in this study, it may be worth investigating in future work. Moreover, Mg(OH)_2_ NPs showed an inhibitory effect on biofilm formation against *S. mutans* and *S. sobrinus*, but to use this material for oral care medicine, investigation of its effect against dental plaque in the oral cavity (i.e., a complex of numerous bacteria) is required. In this study, we have only shown efficacy against cariogenic bacteria, but periodontopathic bacteria and pathogenic fungi are also present in the oral cavity. Since antimicrobial agents are not effective against fungi, it is necessary to develop fungus-specific (highly selective toxicity) agents. Recently, cationic dicephalic surfactants [41], phenylpropenoids (eugenol), and monoterpenes (citral) [42] have been reported as effective anti-biofilm agents for pathogenic fungi (especially *Candida albicans*). If our Mg(OH)_2_ NPs are found to have antifungal effects against *C. albicans*, they can be applied as a broad-spectrum antimicrobial agent. Although there is still much to be explored experimentally, this study showed that Mg(OH)_2_ NPs have potential for use as a caries risk-reducing agent. Although highly effective in inhibiting biofilm formation, it was less effective in dispersing biofilms; therefore, one of its possible uses may be as a sealant to protect tooth surfaces, rather than as an oral care agent such as a mouthwash. The inclusion of nanoparticles in sealants is expected to inhibit the adhesion of caries-causing bacteria. This is a subject for future research, and it is necessary to consider the residence time in the oral cavity and the bond strength with the tooth surface when NPs are incorporated in the sealant. We also propose that one application is to use as a temporary indirect pulp-covering material when the pulp may be exposed to the outside as a result of softened dentin owing to caries.

## 5. Conclusions

In this study, we reported the inhibitory effects of Mg(OH)_2_ NPs of three different sizes (NM80, NM300, and NM700) against cariogenic bacteria, namely *S. mutans* and *S. sobrinus*. All Mg(OH)_2_ NPs inhibited biofilm formation but not bacterial growth, thus posing a low risk of disrupting the oral flora. Larger sized Mg(OH)_2_ NPs were more effective in inhibiting biofilm formation, especially the process of adhesion. The inhibition of caries-causing bacterial biofilms can significantly reduce the risk of dental caries. Mg(OH)_2_ has little effect on the human body and can be used as an oral care product. Therefore, Mg(OH)_2_ NPs have potential applications as dental caries prophylaxis, such as a sealant for pit and fissure sealing or indirect pulp capping during the deep treatment of infected dentin.

## Figures and Tables

**Figure 1 nanomaterials-13-00864-f001:**
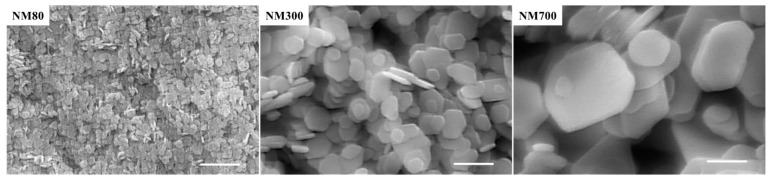
Scanning electron microscopy images of NM80, NM300, and NM700 Mg(OH)_2_ nanoparticles. Scale bars indicate 500 nm (magnification: 40,000×).

**Figure 2 nanomaterials-13-00864-f002:**
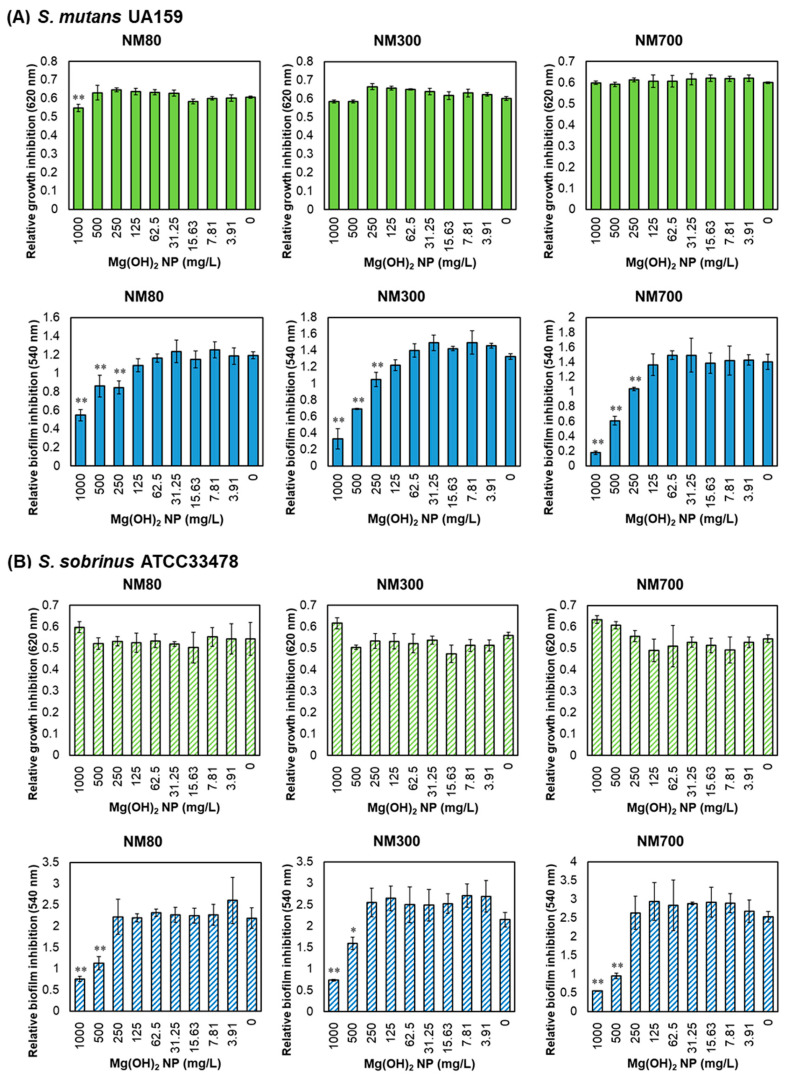
Mg(OH)_2_ NP inhibitory effects on growth and biofilm formation. Relative growth inhibition at an absorbance of 620 nm (green) and relative biofilm inhibition at an absorbance of 540 nm (blue) of *S. mutans* (**A**) and *S. sobrinus* (**B**) were indicated at 1000 mg/L–0 mg/L Mg(OH)_2_ NP (a 2-fold serial dilution was applied). Error bars indicate standard deviations of at least three experiments. Student’s *t*-tests were used to compare the control (0 mg/L) and other groups (* indicates a *p*-value < 0.05 and ** indicates a *p*-value < 0.01).

**Figure 3 nanomaterials-13-00864-f003:**
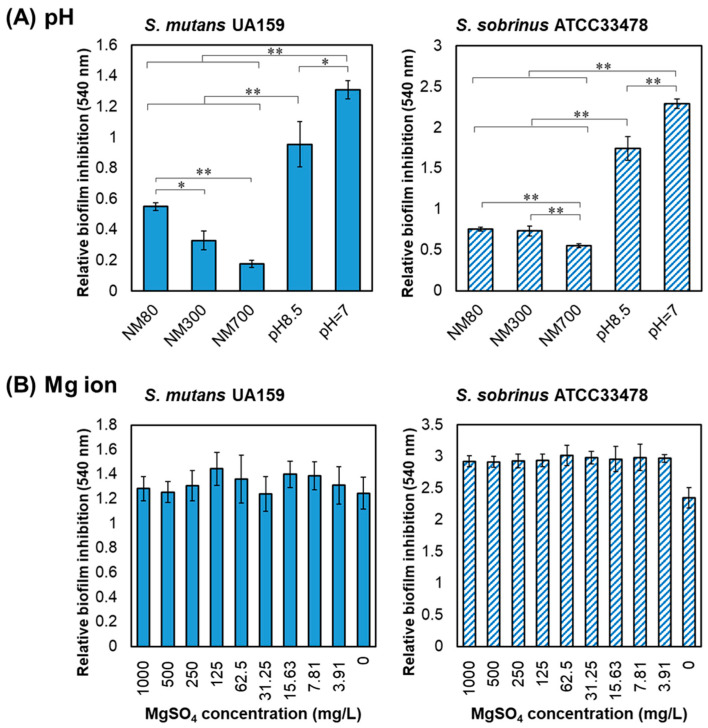
Influence of pH and magnesium ions on biofilm formation. (**A**) Comparing the relative biofilm inhibition of pH 8.5, pH 7, and each Mg(OH)_2_ NP size (1000 mg/L NM80, NM300, and NM700). (**B**) Relative biofilm inhibition by 1000 mg/L–0 mg/L MgSO_4_ (a two-fold serial dilution was applied). Left graphs indicate *S. mutans* and right graphs indicate *S. sobrinus*. Error bars indicate standard deviations of at least three experiments. Student’s *t*-tests were used to compare the two groups (* indicates a *p*-value < 0.05 and ** indicates a *p*-value < 0.01).

**Figure 4 nanomaterials-13-00864-f004:**
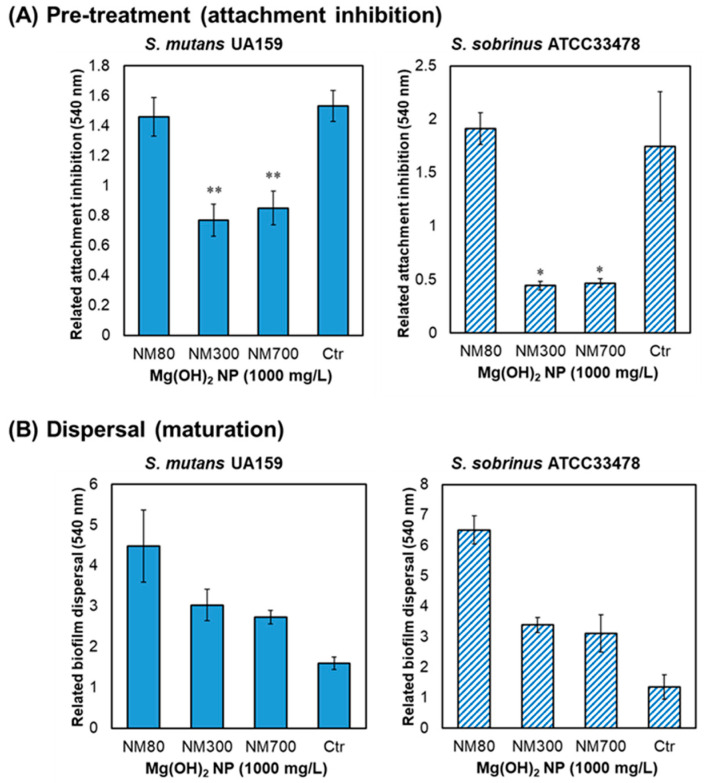
Examination of the biofilm inhibition process by Mg(OH)_2_ NPs. (**A**) Relative bacterial attachment inhibition by Mg(OH)_2_ NP pre-treatment. (**B**) Biofilm dispersal test of Mg(OH)_2_ NPs. All Mg(OH)_2_ NP size groups were used at a concentration of 1000 mg/L. Left graphs indicate *S. mutans* and right graphs indicate *S. sobrinus*. Details of these experiments are indicated in the Experimental section. Error bars indicate standard deviations of at least three experiments. Student’s *t*-tests were used to compare the control (Ctr; PBS) and the other groups (* indicates a *p*-value < 0.05 and ** indicates a *p*-value < 0.01).

## Data Availability

The data presented in this article are available upon request from the corresponding author.

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
