# Peer review of "Magnesium Hydroxide Nanoparticles Inhibit the Biofilm Formation of Cariogenic Microorganisms"

_nanomaterials, 2023, doi:10.3390/nano13050864_

Round 1

Reviewer 1 Report

This study demenstrated the inhibitory effects of magnesium hydroxide nanoparticles of different sizes on the biofilm formation of cariogenic microorganisms, Streptococcus mutans and Streptococcus sobrinus. Overall, the manuscript is well structured and presented. It deserves its publication after some minor revisions.

Methods

1. How can you control the concentration of 5% CO2 during the incubation?

2. How can you generate the sizes of approximately 80, 300, and 700 nm, respectively? How can you make sure they have an equal size at the same scale?

Results

1. L160, magnesium hydroxide ions or magnesium ions? 

2. The Y-axis used in this study is wrong in Figures 2, 3 and 4. It should be relative growth or relative biofilm formation or dispersal.

3. L180-182, what do you mean? Please explain your reaults of Figure 4B in detail.

Discussion

1. L215-216, this conclusion should be narrowed down to your particle size as you just tried three sizes. How about 800 nm, 1000 nm or larger?

2. What do you mean  the accumulation of mature biofilms?

3. L240-242, cite the literature case by case for biofilm control, such as Shahina et al., DOI: 10.1128/spectrum.03183-22; biomolecular interactions and regulation as discussed by Liu et al., DOI: 10.1016/j.biotechadv.2022.107915; or others.

4. L253, it should be discussed how to make your particles well attach on tooth surfaces  when used as a sealant. How about the lifespan of your materials? 

Author Response

Thank you for reviewing our paper and providing us with all of your helpful comments. We revised and updated our paper following your advice, and we responded to your comments one by one in the section below.

Methods

  1. How can you control the concentration of 5% CO2 during the incubation?

→A CO2 incubator was used in all experiments. As you suggested, we indicated this information on line 92.

  1. How can you generate the sizes of approximately 80, 300, and 700 nm, respectively? How can you make sure they have an equal size at the same scale?

→In a previous study, we explained how these nanoparticles were made. The size was controlled by changing the mixing chemical ratio and reaction time. The size measurements were performed by dynamic light scattering (DLS). If we were to describe these methods in further detail in the text, we would duplicate the information in many of our earlier papers. Therefore, we addressed them in this paper by stating the methods briefly and giving the citations.

Results

  1. L160, magnesium hydroxide ions or magnesium ions?

→This was meant to be “magnesium ions”. We revised the word choice (line 170). Thank you for pointing this out.

  1. The Y-axis used in this study is wrong in Figures 2, 3 and 4. It should be relative growth or relative biofilm formation or dispersal.

→As suggested, we revised the Y-axis and the figure legends to include “relative inhibition” (Figure 2, 3, and 4).

  1. L180-182, what do you mean? Please explain your reaults of Figure 4B in detail.

→To determine the effect on the biofilm after maturation, we performed the dispersal experiment on maturated biofilms. If this process was effective, it could be used in mouthwashes. However, no dispersion-removal effect was observed. Rather, there was an increase in apparent biofilm formation compared to the untreated (Ctr) groups in both strains. We revised this section to be clearer (lines 189–196).

Discussion

  1. L215-216, this conclusion should be narrowed down to your particle size as you just tried three sizes. How about 800 nm, 1000 nm or larger?

→In this study, we used nanoparticles with sizes of 80, 300, and 700 nm. As you pointed out, based on the results of this study, we can expect good results for particles 1000 nm and larger. However, we used a maximum size of 700 nm in this study because we thought that nanoparticles larger than 1000 nm would be beyond the scope of “nanoparticles”, and therefore, slightly different from our research basis of nanoparticle effectiveness. In the future, we will consider comparing the effects of larger sizes. We added this discussion in lines 236–240. Thank you for your suggestion.

  1. What do you mean the accumulation of mature biofilms?

→Thank you for identifying that this sentence was difficult to understand. We revised the sentence to “This has not been verified, but this may be due to the deposition of Mg(OH)2 NPs on the mature biofilm.” (line 236).

  1. L240-242, cite the literature case by case for biofilm control, such as Shahina et al., DOI: 10.1128/spectrum.03183-22; biomolecular interactions and regulation as discussed by Liu et al., DOI: 10.1016/j.biotechadv.2022.107915; or others.

→As suggested, we included genetic-related biofilm regulation in the Discussion section (lines 267–271). Although the present report is summarized from the physical perspective of adhesion inhibition, we consider it important to investigate the genetic mechanism in the future. Thank you for your helpful suggestion.

  1. L253, it should be discussed how to make your particles well attach on tooth surfaces when used as a sealant. How about the lifespan of your materials?

→We are currently conducting research on inhibiting the adhesion of cariogenic bacteria by incorporating nanoparticles into sealants. It would be to our disadvantage to reveal the details of this plan in this paper. Thank you for your suggestion and we added a little more discussion about antimicrobial properties and adhesive strength of the sealant (lines 286–289).

Reviewer 2 Report

The paper describes the possibility of using nanoparticles of different sized to avoid the formation of biofilm by oral bacteria thus reducing the risk of developing a caries.Interesting experiments have been done to test the anti-biofilm properties of the NPs but how the authors intent to delivery and maintain the NPs in the mouth and what were the results "in vivo" is missing. So the data presented seem to be one more material with anti-biofilm properties among the big quantity of proposals of this type already published. I dont find data particularly interesting and consider that the application of this NP to avoid the caries is missed.

Author Response

Thank you for your comments. As you pointed out, in vivo testing was not done at this time. This study shows the inhibitory effect of magnesium hydroxide nanoparticles on cariogenic bacteria, which has not been reported before. Furthermore, this paper demonstrates that the inhibitory effect is an inhibition of adhesion. We think this is a useful paper as new knowledge for the prevention of dental caries, one of the world’s most common diseases. As mentioned in the discussion, we envision the use of these NPs in the oral cavity, including their inclusion in sealants and pulp coating materials. We are currently promoting research on this as well, but first we thought we should publicize the excellent adhesion inhibitory effect of magnesium hydroxide nanoparticles and summarize the results at this time.

Reviewer 3 Report

R1

Dear editor and authors!

                The article „Magnesium hydroxide nanoparticles inhibit the biofilm formation of cariogenic microorganisms” concerns the possible use of magnesium hydroxide nanoparticles as anti-caries compounds. The research was carried out with the selection of appropriate Streptococcus strains, however, it could be extended to include anaerobic bacteria. The above studies give a certain indication of the direction of further development of research in the above aspect. In my opinion, the research is interesting, but it should be a they should be extended with modern visual techniques (especially with biofilm).

  I think the article will be to the liking of readers of Nanomaterials, however I consider following improvements necessary before publication: Major Revision

General comments in paragraphs:

1. The introduction needs to be clarified:

40-41: „come into  contact with the hard tissues”.

41-42: „Mutans streptococci are closely related to caries-causing bacteria” Troublesome statement - change it.

TIPS:

·         What is the process of caries formation?

·         What anti-caries materials are currently used?

The introduction in terms of expertise in microbiology is poor. Please correct it.

Materials and Methods

79   „Bacteria cultivation” better -> Bacterial strains

80: Check number of strains „ATCC33478”/ space!

·         The CV method is acceptable, but it is best to extend it with (Syto9/PI).

The work requires correction of English.

Discussion extras:

Please consider expanding discussions or introduction with new interesting information. They will help explain the mechanism of action of the compounds and will bring very interesting information to the discussion:

1. https://doi.org/10.3390/ijms23031533  (materials in potential  anticaries activity)
2. https://www.nature.com/articles/s41598-021-88244-1 (biofilm methods and visualitaton)

Maybe these works will help.
Regards  R1

Author Response

Thank you for reviewing our paper and providing us with your helpful comments. We revised and updated our paper following your advice, and we responded to your comments one by one in the section below.

General comments in paragraphs:

  1. The introduction needs to be clarified:

40-41: „come into  contact with the hard tissues”.

41-42: „Mutans streptococci are closely related to caries-causing bacteria” Troublesome statement - change it.

TIPS:

What is the process of caries formation?

What anti-caries materials are currently used?

The introduction in terms of expertise in microbiology is poor. Please correct it.

→As suggested, we revised the introduction. We have also corrected any misleading wording as you have pointed out. The mechanism of caries formation is explained a little more clearly (lines 40–55). Currently, the only treatment for dental caries is to essentially scrape off the affected area. Since there is no generalized prophylactic drug, dental caries is still one of the most common diseases in the world. For this reason, we have been researching and proposing materials that can help prevent dental caries.

Materials and Methods

79   „Bacteria cultivation” better -> Bacterial strains

→As suggested, we revised this (line 89).

80: Check number of strains „ATCC33478”/ space!

→As suggested, we revised this (line 90).

The CV method is acceptable, but it is best to extend it with (Syto9/PI).

→Thank you for your suggestion. In this study, we focused on biofilm formation inhibition. Therefore, we thought that dead bacteria or bacterial arrival were not so important. Our Mg(OH)2 NPs did not inhibit growth but inhibited biofilm formation. This means that these NPs have the potential to reduce the risk of dental caries without altering the oral flora (we discussed this in Discussion section lines 246–253). However, as you suggested, microscopic images of biofilms using Syto9 can provide visual clarity. We will introduce this into our future experimental strategy.

The work requires correction of English.

→As you suggested, our paper was proofread by a native speaker.

Discussion extras:

Please consider expanding discussions or introduction with new interesting information. They will help explain the mechanism of action of the compounds and will bring very interesting information to the discussion:

  1. https://doi.org/10.3390/ijms23031533 (materials in potential anticaries activity)
  2. https://www.nature.com/articles/s41598-021-88244-1 (biofilm methods and visualitaton)

→Thank you for introducing these two very interesting papers. We have included them in our discussion (lines 267–271). Thanks to your advice, the quality of our paper was improved.

Round 2

Reviewer 2 Report

The response of the authors to my most important concerns do not add any interesting data and did not solve the specific points I highlighted. I have been looking at recent reviews on the topic and I confirm that there is a lot of materials and NPs that have been proved to avoid oral bacteria biofilm formation or reducing the adhesion of bacteria and even having antibacterial properties. Still some concerns have been detected as regards the residence time of NPs in the oral environment and the way to delivery efficiently the NPs to the tooth. The study I revised does not give any contribution to analyze/solve these problems. For this reason I consider the paper have a low level of interest for publication.

Author Response

Thank you very much for your comments again.

As suggested, we have also confirmed the existence of NPs with various antimicrobial properties, which are discussed in the Introduction (line 61) and Discussion (line 254). The important points of this study are (1) magnesium hydroxide nanoparticles, which have less adverse effects on the human body among metal nanoparticles, are particularly effective in inhibiting caries-causing bacterial biofilms, (2) they are effective in inhibiting adhesion during the biofilm formation process, (3) the larger the particle size, the higher the inhibitory effect, and (4) they do not affect growth and thus have a low risk of disturbing flora, a novel finding not reported previously. The method of delivery and residence time you mention is not the main topic of this paper, but is a subject for future work. One of the best possible solutions to these concerns of yours is to incorporate them into a sealant. Our magnesium hydroxide nanoparticles are highly effective at inhibiting adhesion, and we believe that sealants are the simplest and most useful way to maintain them in the mouth for an extended period of time. We are currently working on this and will provide a description in the Discussion (line 287). We recognize that what you mention is very important, and we believe that we need to move our research closer to the clinic in the future. At this point, we have decided that we should summarize our research as a basic research phase and have submitted this paper. We appreciate your valuable comments very much.

Reviewer 3 Report

The manuscript improved significantly after the corrections were made. The authors have followed the recommendations of the reviewers.

Recommends publications in their current form.

Author Response

Thank you very much for all your constructive and professional comments.